Spatial modeling of long-term air temperatures for sustainability: evolutionary fuzzy approach and neuro-fuzzy methods

Sadeghi-Niaraki Abolghasem a.sadeghi.ni@gmail.com 1 2
Kisi Ozgur 3
Choi Soo-Mi 2
1 Geoinformation Tech. Center of Excellence, Faculty of Geomatics Engineering, K.N. Toosi University of Technology , Tehran , Iran
2 Department of Computer Science and Engineering, Sejong University , Seoul , South Korea
3 Faculty of Natural Sciences and Engineering, Ilia State University , Tbilisi , Georgia
Jones Roger
Electronic publication date: 2020 Aug 14
Publication date: 2020
Volume: 8
Electronic Location ID: e8882
Received 2019 Jun 27; Accepted 2020 Mar 10
Copyright: ©2020 Sadeghi-Niaraki et al.
Copyright year: 2020
Copyright holder: Sadeghi-Niaraki et al.
License: This is an open access article distributed under the terms of the Creative Commons Attribution License, which permits unrestricted use, distribution, reproduction and adaptation in any medium and for any purpose provided that it is properly attributed. For attribution, the original author(s), title, publication source (PeerJ) and either DOI or URL of the article must be cited.
License URL: https://creativecommons.org/licenses/by/4.0/

Keywords: Long-term air temperature, Estimation, Evolutionary fuzzy, Neuro-fuzzy, GIS

Funding: Ministry of Science and ICT Information Technology Research Center IITP-2020-2016-0-00312 This research was supported by the MSIT (Ministry of Science and ICT), Korea, under the ITRC (Information Technology Research Center) support program (IITP-2020-2016-0-00312). The funders had no role in study design, data collection and analysis, decision to publish, or preparation of the manuscript.

==============================
This paper investigates the capabilities of the evolutionary fuzzy genetic (FG) approach and compares it with three neuro-fuzzy methods—neuro-fuzzy with grid partitioning (ANFIS-GP), neuro-fuzzy with subtractive clustering (ANFIS-SC), and neuro-fuzzy with fuzzy c-means clustering (ANFIS-FCM)—in terms of modeling long-term air temperatures for sustainability based on geographical information. In this regard, to estimate long-term air temperatures for a 40-year (1970–2011) period, the models were developed using data for the month of the year, latitude, longitude, and altitude obtained from 71 stations in Turkey. The models were evaluated with respect to mean absolute error (MAE), root mean square error (RMSE), Nash–Sutcliffe efficiency (NSE), and the determination coefficient (R2). All data were divided into three parts and every model was tested on each. The FG approach outperformed the other models, enhancing the MAE, RMSE, NSE, and R2 of the ANFIS-GP model, which yielded the highest accuracy among the neuro-fuzzy models by 20%, 30%, and 4%, respectively. A geographical information system was used to obtain temperature maps using estimates of the optimal models, and the results of the model were assessed using it.

Introduction

Regional climate is critical to the efficiency of crops as it deeply influences their yield potential. Rainfall, solar irradiance, and the temperature of the air are factors affecting crop yield, improvement, and growth (Ali-Nezhad & Eskandari, 2012). With regard to the phonological levels of the ripening of a crop, temperature is considered vital (Rosenzweig & Liverman, 1992). Moreover, crop growth is done within only specific temperature ranges. While the growth of a plant relies heavily on temperature, the plant species is the most significant determinant of the range of temperature ideal for its growth (Hasanuzzaman, Nahar & Fujita, 2013). Temperature determines the crop’s survival and growth in each region (Cobaner et al., 2014). Brunetti et al. (2014) discuss the pros and cons of three spatial models such as multi-linear regression with local improvements (MLRLI), regression kriging (RK), local weighted linear regression (LWLR) considering elevation input for temperature assessment in Italy. Gonzalez-Hidalgo et al. (2015) employ spatial trend analysis based on the interpolation method such as radial weight with a Gaussian shape using angular weight for temperature data analysis. Serrano-Notivoli, Beguería & De Luis (2019) evaluate spatial, temporal variability of temperature using methods such as Generalized Linear Mixed (GLMM) and Generalized Linear Models (GLM) with the inclusion of spatial predictors such as latitude, longitude, altitude, and distance.

Techniques from artificial intelligence (AI), such as the adaptive neuro-fuzzy inference system (ANFIS) and artificial neural networks (ANN), have been used in such subjects as agro-hydrology, agro-meteorology, and engineering for water resources. This study focuses on the relevant literature. Zhu et al. (2019) proposed a model using different machine learning techniques such as multilayer perceptron neural network models (MLPNN), adaptive neuro-fuzzy inference systems (ANFIS) with fuzzy c-mean clustering algorithm (ANFIS_FC), ANFIS with grid partition method (ANFIS_GP), and ANFIS with subtractive clustering method (ANFIS_SC) for prediction of water temperature. Khosravi et al. (2018) employs a model using multilayer feed-forward neural network (MLFFNN), radial basis function neural network (RBFNN), support vector regression (SVR), fuzzy inference system (FIS) and adaptive neuro-fuzzy inference system (ANFIS) for solar radiation prediction.

The abductive neural network method was utilized by Abdel-Aal (2004) to predict air temperature every hour. Using data concerning seasonality and changing the parameters of an ANN model, Smith, McClendon & Hoogenboom (2005) developed an ANN for temperature forecasting. Dew-point temperature was modeled by Shank, Hoogenboom & McClendon (2008) using neural networks, and Turkey’s long-term temperature forecasting was applied by Bilgili & Sahin (2009) to each month. A new method utilizing the Yule–Walker equation and ANNs was proposed by Chattopadhyay, Jhajharia & Chattopadhyay (2011) to model the time series of monthly maximum temperatures in northeast India. Turkey’s mean air temperature for each month was modeled by Şahin (2012) using ANN and remote sensing. By using geographical data, the performance of ANNs and ANFIS was compared in terms of forecasting air temperature in Iran by Kisi and Shiri. In another study, the monthly mean of Turkey’s air temperature was predicted by Cobaner et al. (2014) using ANNs and ANFIS. All past assessments have used ANFIS and ANNs, and none of them has evaluated fuzzy methods for long-term modeling of temperature by applying geographical information. Thus, this paper aims to model long-term temperatures using geographical data and compare an evolutionary fuzzy method with three approaches based on ANFIS.

Some researchers in climatology, biogeography, hydrology, meteorological, agriculture, and ecology have applied temperature modeling using spatial/geographical information systems (GIS) (Ninyerola, Pons & Roure, 2000). Goodale, Aber & Ollinger (1998) applied predictors to interpolate temperature and precipitation in Ireland using spatial information (latitude, longitude, and altitude). Geostatistical model kriging was employed by Benavides et al. (2007) to model temperature. Chuanyan, Zhongren & Guodong (2005) compared the results of conventional predictive models, such as linear regression and geostatistical interpolation techniques (e.g., ordinary kriging, splines, and inverse distance weight), in terms of spatial distribution in modeling surface air temperature. Ninyerola, Pons & Roure (2000) also employed GIS-based techniques for the accurate prediction and mapping of the nonlinear behavior of air temperature over space and time.

To predict long-term monthly air temperature with spatial data for achieving sustainability, in this research the evolutionary fuzzy genetic (FG) model applicability is examined. The results of the proposed model were evaluated by comparing with those of three ANFIS models: ANFIS using fuzzy c-means clustering (ANFIS-FCM), ANFIS using grid partitioning (ANFIS-GP), and ANFIS using subtractive clustering (ANFIS-SC). In this regard, training and testing the FG and ANFIS models were conducted on 71 weather stations data in Turkey. The results were also evaluated using the GIS method.

Study Area

The 37th largest country in the world, Turkey is located at 38.9637° N latitude and 35.2433° E longitude with an area of 780,000 km2, 25 million hectares of which is suited for agriculture. Turkey is a mountainous country with large plains. The highest mountain, Ararat, is 5165 m high. The distribution of temperature and weather patterns are significantly influenced by the orientation and elevation of mountains in the country.

This study used monthly mean temperatures gathered from 71 stations of the Turkish State Meteorological Service. This information included a 40-year dataset from 1970–2011 for each station. Table 1 shows the temperature (∘C), longitude, latitude, and altitude above sea level as recorded by each station. The temperature varied from −11.5 °C in January (Ardahan) to 32 °C in July (Urfa) in the long term. The high variation may have occurred because of the sea rounding these areas, the high mountains along the coasts of the Black Sea and the Mediterranean Sea, and the mountainous eastern Anatolia region (Cobaner et al., 2014). Figure 1 shows the meteorological stations distribution in Turkey.

Methods

Fuzzy approach

Fuzzy logic has been used in such fields as business, engineering, and the sciences (Zadeh, 1965). Figure 2 shows units of fuzzification of a general fuzzy system including a fuzzy inference engine, a fuzzy rule base, and defuzzification. Fuzzy logic is based on the idea of using hybrid systems, where part of one system and part of another are used instead of only one. The degree of dependence on a system is a number between zero and one.

Table 1 Summary of the geographical information of the studied weather stations (Kisi & Shiri, 2014).

Station	Lat.
(°N)	Long.
(°W)	Alt.
(m)	∗T (°C)	Station	Lat.
(°N)	Long.
(°W)	Alt.
(m)	∗T (°C)	
Adana	37	35.32	27	19.12	K.maras	37.3	40.77	572	16.81	
Adiyaman	37.76	38.27	672	17.24	Karabuk	41.2	32.62	259	13.33	
Afyon	38.74	30.55	1,034	11.28	Karaman	37.17	33.22	1,025	11.88	
Agri	39.72	43.05	1,632	6.18	Kars	40.59	43.08	400	4.81	
Aksaray	38.37	34.03	965	12.02	Kastamonu	41.37	33.77	800	9.73	
Amasya	40.65	35.83	412	13.63	Kayseri	38.72	35.49	1,093	10.44	
Ankara	39.97	32.86	891	11.98	Kirikkale	39.84	33.52	748	12.57	
Antalya	36.9	30.79	54	18.40	Kirklareli	41.74	27.22	232	13.20	
Ardahan	41.11	42.7	1,829	3.75	Kirsehir	39.16	34.15	1,007	11.35	
Artvin	41.18	41.82	628	12.03	Kilis	36.72	37.12	638	17.10	
Aydin	37.84	27.84	56	17.63	Konya	37.99	32.56	1,031	11.58	
Balikesir	39.63	27.92	102	14.55	Kutayha	39.42	29.99	969	10.75	
Bartin	41.62	32.35	30	12.63	Malatya	38.35	38.31	948	13.74	
Batman	37.89	41.12	540	16.51	Manisa	38.61	37.4	71	16.98	
Bayburt	40.25	40.43	1,584	6.93	Mardin	37.31	40.73	1,040	16.15	
Bilecik	40.14	29.97	539	12.50	Mugla	37.29	28.37	646	15.01	
Bingol	38.88	40.49	11.77	11.95	Mus	38.74	41.49	1,320	9.71	
Bitlis	38.39	42.12	1,573	9.53	Nigde	37.97	34.69	1,211	11.08	
Bolu	40.74	31.6	743	10.53	Osmaniye	37.1	36.25	94	18.24	
Burdur	37.72	30.29	967	13.18	Rize	41.04	40.5	9	14.23	
Bursa	40.23	29.01	100	14.56	Sakarya	40.77	30.39	31	14.44	
Canakkale	40.15	26.41	6	15.01	Samsun	41.35	36.24	4	14.38	
Cankiri	40.61	33.61	751	11.18	Siirt	37.93	41.94	896	16.11	
Corum	40.54	34.94	776	10.57	Sinop	42.02	35.15	32	14.09	
Denizli	37.76	29.09	425	16.21	Sivas	39.74	37.02	1,285	9.08	
Elazig	38.67	39.22	990	13.03	S.urfa	37.16	38.79	549	18.38	
Erzincan	39.74	39.5	1,218	10.89	Sirnak	37.52	42.45	1,350	14.13	
Erzurum	39.95	41.17	17.58	5.40	Tekirdag	40.99	27.49	3	13.96	
Eskisehir	39.78	30.58	786	10.84	Tokat	40.3	36.56	608	12.48	
Gaziantep	37.07	37.39	855	15.07	Trabzon	41	39.78	39	14.68	
Giresun	40.92	38.39	37	14.53	Tunceli	39.1	39.55	978	12.83	
Gumushane	40.46	39.47	1219	9.55	Usak	38.67	29.4	919	12.51	
Hakkari	37.57	43.75	1,728	10.26	Van	38.49	43.39	1,661	9.24	
Hatay	36.36	36.28	82	18.26	Yalova	40.66	29.27	4	14.67	
Igdir	39.92	44.06	858	12.08	Yozgat	39.82	34.81	1,298	8.97	
Istanbul	40.98	28.82	33	15.01						

In the inference mode, the fuzzy system takes inputs and outputs data. By using a map of association, namely fuzzy associative memory, the inputs are transformed into the equivalent output sets by fuzzy system learning (Kosko & Toms, 1993). Some “black box” approaches such as ANNs can function, for example, in regression, but fuzzy systems are clearer and more pliable. Thus, it is clear how fuzzy systems perform and modify processes (Russell & Campbell, 1996).

The fuzzy method used here is as follows. Several subsets using Gaussian member function are formed from the input and output parameters. ck fuzzy rules are available, where c and k are the subsets and inputs numbers, respectively. Efficiency increases with the number of subsets, and the rule base increases in weight, rendering construction more complex. In the case of only one input, x, and k subsets, the rule base yields yn (n = 1, 2, …, k2) (Şen, 1998).

Figure 1 Meteorological stations distribution in Turkey (the map was provided by the State State Hydraulic Works (DSI), see http://en.dsi.gov.tr/about-dsi-)

In the case of one variable input, x with four values of “low,” “medium,” “high,” and “very high,” four rules are possible:

R1: IF x is “low,” THEN y1

R2: IF x is “medium,” THEN y2

R3: IF x is “high,” THEN y3

R4: IF x is “very high,” THEN y4

The output is single weighted, y, as a weighted average of the outputs of these four rules as follows: (1) y=∑n=14wn⋅yn ∑n=14wn

where the degree of membership, wn, is investigated for each x to be assigned an equivalent yn after triggering each rule.

Figure 2 A typical fuzzy inference system.

Therefore, having formed the rule base, any assortments of subsets of the parameters used as input to the fuzzy system, Eq. (1), can be used to compute the output values (y) (Şen, 1998).

The proposed fuzzy base rule can be computed using the datasets used as input and output as follows:

1. using the smallest number of inputs;

2. assigning a specific membership function to each input;

3. computing the value of membership (wn) of x in all fuzzy subsets;

4. calculating the output yn simultaneously with the weight set wn;

5. renewing all other data points; and

6. calculating the weighted average using Eq. (1) (Kiszka, Kochanskia & Sliwiska, 1985a;Kiszka, Kochanskia & Sliwinska, 1985b).

As any change in the subsets makes a direct effect on the functionality of the fuzzy model, forming the fuzzy subsets is among the most challenging problems in the area. It is thus vital to define the membership functions optimally to maximize modeling efficiency (Kisi & Cengiz, 2013). The optimal membership functions are defined in this research using the genetic algorithm.

Genetic algorithm

GAs have been researched in engineering since 1960, when they were introduced by Holland (Abraham & Jain, 2005; Ortiz Jr et al., 2004). In general, GAs try to imitate the Darwinian concept of natural selection. A GA first generates a set of possible solutions and tries to find the best approach to survival to form a new population of solutions, which assesses the real solution better than before. The challenge of creating better and fitter solution sets is the basic principle of GAs. They are utilized to find the best and most optimal solutions to solve difficult problems. Utilizing this approach provides a useful option for solutions (Aijun et al., 2004; Tsai, Liu & Chou, 2004). While other methods are restricted due to their suppositions, GAs have fewer such limits (Jean, Lin & Chou, 2007).

Adaptive neuro-fuzzy inference system

ANFIS, a general investigation tool with the quality to approximate continuous real functions on compact sets, was developed in 1993 by Jang. In its structure, nodes are attached to one another by directional links. A node function containing variable or constant parameters introduces the node (Jang, Sun & Mizutani, 1997).

The example below presents a typical fuzzy inference system (FIS) with an output f, three inputs x, y, and z, and two if–then fuzzy rules of the Takagi and Sugeno type: (2) Rule 1: If x is A1, y is B1, and z is C1, thenf1=p1x+q1y+r1z+s1

(3) Rule 2: If x is A2, y is B2, and z is C2, thenf2=p2x+q2y+r2z+s2

where f1 and f2 are the output functions of rules 1 and 2, respectively. Figure 3 shows the structure of ANFIS. The functions of the nodes are as follows.

Figure 3 ANFIS model for temperature prediction.

Ol,i = ϕAi(x) indicates the node function for each square node i of layer 1 that is adaptive, and i = 1, 2, x, is the ith input node; Ai is a linguistic label (i.e., “small” or “big”) for this node function. When input x satisfies quantifier Ai, Ol, i, is the membership function (MF) of fuzzy set A (e.g., A1, A2, B1, B2, C1, C2) and indicates its degree. ϕAi(x) is mainly a Gaussian function ranging between zero and one as the minimum and maximum levels, respectively: (4) ϕAix= exp−x−aibi2

where {ai, bi} is the parameter set. These layer parameters are considered to be assumed. The circle nodes of layer 2 have index Π, which indicates the multiplication of the inputs and the sending of the product. For instance, wi = ϕAi(x)ϕBi(y)ϕCi(z), i = 1, 2. The output of each node shows the impression level of a rule. Circle nodes of layer 3 take the label N. Following this, the ratio of the level of impression of the ith rule on the sum of the levels of all rules is computed by the ith node using (5) wi=wiw1+w2,i=1,2.

The node function of the square nodes of layer 4 is O4,i=w ¯ifi=w ¯ipix+qiy+riz+si, where w ¯i represents the layer 3 output and the parameter series is pi,qi,ri,si. These layer parameters are called consequent parameters. The single circle nodes of layer 5, labeled “Σ,” sums all incoming signals and returns the result as the final output: (6) O5,i= ∑i=1w ¯ifi=∑iwifi ∑iwi.

The ANFIS network acts similarly to a first-order Sugeno FIS (Kisi, 2015). Linear or fixed-valued functions are used in ANFIS to generate the output. Detailed information concerning ANFIS is available in Jang’s study (1993).

Grid partitioning

This approach provides independent partitions of prior variables (Jang, 1993). The membership functions of all prior variables can be determined by prior knowledge and experience. Expressing the meaning of the linguistic terms of a context is the goal of designing these partitions. However, in many systems, there is no specific information accessible to these partitions. Thus, the ranges of the prior variables may easily be divided into membership functions that are equally shaped and spaced. The membership functions may be located suitably in case the system’s input–output data are available. The rule base should be generated in a manner that perfectly covers previous fuzzy set combinations. The membership functions of each variable are built apart from those of others, and this is the main disadvantage of this method as this causes it to overlook the relationship among variables (Vernieuwe et al., 2005).

Subtractive clustering

The ANFIS subtractive clustering (ANFIS-SC) model is defined by merging ANFIS with subtractive clustering. In this model, a possible cluster center is each data point, which is not a grid point (Chiu, 1994). It is thus an extended model of Yager & Filev (1994), the mountain clustering method.

In this approach, the number of effective “grid points” that should be investigated are similar to the number of data points independently of the number of dimensions of the problem. One of the advantages of this approach is that a tradeoff between computational complexity and accuracy is unnecessary because the need to define a grid resolution is removed. The scale of the mountain method to accept or reject the cluster centers is also extended in subtractive clustering.

The impressive radius is vital to defining the number of clusters. Too many smaller clusters require more rules if a small radius is chosen, and vice versa. Thus, selecting a suitable impressive radius for data space clustering is crucial. Defining the number of fuzzy rules and presumptions of the fuzzy MF is the next stage. In the end, the results in the output MF are generated by utilizing linear least squares, which builds a valid FIS (Cobaner, 2011; Sanikhani & Kisi, 2012).

Fuzzy c-means clustering

C-means fuzzy clustering is a form of flexible clustering model in which the data points are combined by calculating possible data points in the feature space. In this model, the mountain clustering approach can be used to calculate the number of clusters and cluster centers (Chiu, 1994; Cobaner, 2011). This model is based on the k-means algorithm, which is unsuitable for big datasets. C-means fuzzy clustering reduces the intra-cluster variance to a minimum (Ayvaz, Karahan & Aral, 2007) and combines data using its clustering algorithm. The c-means fuzzy clustering minimizes either the distance or the objective squared error function (Kisi, 2015).

Spatio–temporal modeling

The use of GIS for the spatial modeling of many real-world phenomena toward implementation for sustainability (Childers et al., 2015; Lundgren & Kjellstrom, 2013) has attracted research attention. Spatial modelling acts like a critical tool to examine the nature or property of real-world phenomena whether being sustained or not. Much of real-world phenomena are dynamic, and involve spatial and temporal changes. Thus, the use of a GIS for temporal and spatial modeling plays an important role in improving model visualization and tracking how much far from sustainability indicators (Fig. 4). The GIS is used for various purposes, such as creating the required spatial data (geographical location) used as input for the prediction model. Using the spatial functions generates a map related to the modeling phenomena to assess their behavior and at different locations. Such spatial statistical functions as zonal statistics assess variation in phenomena as well as minimum and maximum mean values at different locations. A geospatial function, such as the interpolation method, enables the elimination of possible model defects (e.g., data loss in some areas). Geostatistical analysis, such as trend analysis functions, provides forecasting trends and patterns of phenomena.

Figure 4 Spatial temporal modeling.

Model evaluation statistics

Four statistics were used to evaluate the models: mean absolute error (MAE), root mean square error (RMSE), Nash–Sutcliffe efficiency (NSE), and the determination coefficient (R2). RMSE, MAE, and NSE can be expressed as

(7) RMSE=1n∑i=1nATM,i−ATo,i2

(8) MAE=1n∑i=1nATM,i−ATo,i

(9) NSE=1−∑i=1nATM,i−ATo,i∑i=1nATo,i−ATo¯

where ATM and ATo are modeled and observed air temperatures, respectively, n is the number of time steps, and ATo is the mean observed air temperature.

Application and Results

Modeling long-term air temperatures using evolutionary fuzzy and neuro-fuzzy approaches

Four hybrid fuzzy methods—evolutionary FG, ANFIS-GP, ANFIS-SC, and ANFIS-FCM—were compared in terms of predicting long-term air temperatures. The inputs of the models were latitude, longitude, altitude and month of the year. To train and test models, the data of 71 weather stations in Turkey were used. The entire dataset (the 70 stations ×12 months = 840 data items) was divided into three subsets. The first 23 stations were used in the first part, the second set of 23 for the second, and the other 24 stations were used for the last part. In the first application, training and testing were conducted using the first two parts (subsets) and the third part respectively; it was called model 1 (M1). In the second application, training was conducted using the second and third parts and testing was conducted using the first part, and was called model 2 (M2). In the last application, training was conducted using the first and third parts the testing was conducted using the second part, and was called model 3 (M3).

ANFIS-GP was applied to the datasets, and two and three Gaussian membership functions (MFs) were used to find the optimal one. More than three MFs led to worse results and memory-related problems. The number of iterations was varied from 10 to 100 with increments of 10 for each number of MFs. Different values of the radius were used (from 0.1 to 1 in increments of 0.1) for the ANFIS-SC model, and different numbers of clusters were used (from two to eight, incremented by one) for the ANFIS-FCM model. The numbers of iterations used were similar to those in the ANFIS-GP and ANFIS-SC methods. To test the FG models, the optimal number of MFs for the ANFIS-GP models were used. A total of 1,000 iterations and 100 members were used for the FG models. The optimal parameters and structures of the applied models are listed in Table 2. For example, 3 − 3 − 2 − 3, 100 indicates an ANFIS-GP model with three, three, two, and three MFs for the four inputs (month of the year, latitude, longitude, and altitude) over 100 iterations. The number 1,100 indicates the value of the radii and the number of iterations of the ANFIS-SC model, the numbers of clusters and iterations of the ANFIS-FCM model were 5,100. “ 3 − 3 − 2 − 3 , 1,000, 100,” indicating an FG model with three, three, two, and three MFs for the four inputs with 1,000 generations and a population of 100.

Table 2 Optimal parameters and structures of the ANFIS-GP, ANFIS-SC, ANFIS-FCM and FG models.

Model	ANFIS-GP	ANFIS-SC	ANFIS-FCM	FG	
M1	3-3-2-3, 100	1, 100	5, 100	3-3-2-3, 1,000, 100	
M2	3-2-2-2, 100	1, 100	8, 90	3-2-2-2, 1,000, 100	
M3	3-3-2-3, 100	1, 100	7, 100	3-3-2-3, 1,000, 100	

Table 3 specifies the results of training and testing the models in terms of RMSE, MAE, NSE, and R2. In the training stage, the RMSE of the FG model ranged from 1.59 to 1.89, whereas that of ANFIS-GP, ANFIS-SC, and ANFIS-FCM was in the range 2.16–2.95, 1.14–1.45, and 6.13–6.93, respectively. Similar ranges were observed for the MAE, NSE, and R2. The ranges and means reveal that the ANFIS-SC was the most accurate in terms of approximating long-term air temperatures followed by the FG. ANFIS-FCM yielded the worst results in this stage. In the test stage, however, the ranges of RMSE, MAE, NSE, and R2 of the FG models were 1.63–3.26, 1.23–2.08, 0.868–0.967, and 0.882–0.967, respectively, which were superior to the those of the other models. The average RMSE, MAE, NSE, and R2 of the optimal FG models (2.32, 1.59, 0.925, and 0.935) were also higher than those of the ANFIS-GP (2.91, 2.28, 0.890, 0.898), ANFIS-SC (5.41, 3.42, 0.539, 0.768), and ANFIS-FCM (12.5, 8.32, 0.239, 0.250) models. Among the neuro-fuzzy models, ANFIS-GP obtained the highest accuracy. In terms of RMSE, MAE, NSE, and R2, the FG model enhanced the accuracy of ANFIS-GP by 20%, 30%, 4%, and 4%, respectively.

Table 3 Training and testing results of the FG, ANFIS-GP, ANFIS-SC and ANFIS-FCM models.

Process		FG	ANFIS-GP	ANFIS-SC	ANFIS-FCM	
	Model	RMSE (°C)	MAE (°C)	NSE	R2	RMSE (°C)	MAE (°C)	NSE	R2	RMSE (°C)	AE (°C)	NSE	R2	RMSE (°C)	MAE (°C)	NSE	R2	
Training	M1	1.89	1.31	0.955	0.956	2.47	1.98	0.924	0.924	1.24	0.95	0.981	0.981	6.66	5.48	0.448	0.448	
M2	2.19	1.43	0.937	0.938	2.95	2.26	0.886	0.886	1.45	1.10	0.972	0.972	6.93	5.81	0.373	0.373	
M3	1.59	1.25	0.967	0.967	2.16	1.83	0.938	0.938	1.14	0.87	0.983	0.983	6.13	5.06	0.504	0.504	
Mean		1.89	1.33	0.953	0.954	2.53	2.02	0.916	0.916	1.28	0.973	0.979	0.979	6.57	5.45	0.442	0.442	
Testing	M1	2.08	1.45	0.940	0.955	2.74	2.26	0.896	0.904	3.93	2.52	0.78	0.820	7.53	6.17	0.209	0.253	
M2	1.63	1.23	0.967	0.967	2.68	2.10	0.910	0.912	9.09	5.64	−0.036	0.602	22.3	12.5	5.22-	0.119	
M3	3.26	2.08	0.868	0.882	3.31	2.47	0.864	0.878	3.21	2.09	0.872	0.882	7.67	6.30	0.269	0.379	
	Mean	2.32	1.59	0.925	0.935	2.91	2.28	0.890	0.898	5.41	3.42	0.539	0.768	12.5	8.32	0.239	0.250	

A graphical comparison of the four methods is provided in Fig. 5–Fig. 7 for M1, M2, and M3. The obvious finding from Fig. 4 is that the estimates of the FG model were closer to the corresponding observed temperatures than those of the ANFIS-GP, ANFIS-SC, and ANFIS-FCM models. Considerable under/over-estimations were observed in the ANFIS-FCM model. The scatterplots show that the fit line equation of the FG model was y = 0.9910x–0.906, closer to the exact line (y = x) with a higher R2 than the other three models. ANFIS-GP was better than the other two neuro-fuzzy models. Similar trends were seen for M2 and M3 (see Figs. 6 and 7).

Figure 5 Observed and predicted air temperatures by FG, ANFIS-GP, ANFIS-SC and ANFIS-FCM models –M1 models; time variation graphs of (A) FG, (B) ANFIS-GP, (C) ANFIS-SC and (D) ANFIS-FCM and scatterplots of (E) FG, (F) ANFIS-GP, (G) ANFIS-SC and (H) ANFIS-FCM.

Figure 6 Observed and predicted air temperatures by FG, ANFIS-GP, ANFIS-SC and ANFIS-FCM models –M2 models; time variation graphs of (A) FG, (B) ANFIS-GP, (C) ANFIS-SC and (D) ANFIS-FCM and scatterplots of (E) FG, (F) ANFIS-GP, (G) ANFIS-SC and (H) ANFIS-FCM.

Figure 7 Observed and predicted air temperatures by FG, ANFIS-GP, ANFIS-SC and ANFIS-FCM models –M3 models; time variation graphs of (A) FG, (B) ANFIS-GP, (C) ANFIS-SC and (D) ANFIS-FCM and scatterplots of (E) FG, (F) ANFIS-GP, (G) ANFIS-SC and (h) ANFIS-FCM.

Spatial modeling of long-term air temperatures using GIS

Air temperature is a spatio-temporal phenomenon, and thus changes with time and place. Based on their capabilities, GIS spatial functions can be used to improve the model. The GIS can be used to provide additional input for air temperature modeling in the form of spatial data (latitude and longitude). This spatial modeling leads to greater flexibility in the proposed model and renders it more adaptable for practical use. Moreover, the GIS is used to create temperature map-related locations to enable better visualization and interpretation of the outputs.

A classified temperature mapping enables a comparison of the variation in temperature ranges of all methods with respect to M1 (Fig. 8), M2 (Fig. 9), and M3 (Fig. 10). The lowest temperature ranges according to the classified temperature map in M1 were ANFIS-SC: 4.52–8.74, ANFIS-FCM: 6.14–8.95, ANFIS-GP: 8.45–10.22, and FG-FUZZY-G: 8.94–10.47. The maximum temperatures were ANFIS-SC: 18.82–22.75, ANFIS-FCM: 17.45–20.21, ANFIS-GP: 16.02–18.05, and FG-FUZZY-G: 16.75–18.47. The lowest temperatures ranges according to M2 were ANFIS-SC: −5.09–-3.33, ANFIS-FCM: −85.50–-6, ANFIS-GP: 7.22–9.34, and FG-FUZZY-G: 4.46–7.62, and the maximum temperatures were ANFIS-SC: 37.08–-48.67, ANFIS-FCM: 13.38–24.13, ANFIS-GP: 16.35–18.47, and FG-FUZZY-G: 15.21–17.51. The lowest temperature ranges in M3 were ANFIS-SC: 8.15–10.75, ANFIS-FCM: 4.60–9.20, ANFIS-GP: 10.23–11.75, and FG-FUZZY-G: 7.96–10.29), and the maximum temperatures were ANFIS-SC: 18.08–19.98, ANFIS-FCM: 23.60–26.73, ANFIS-GP: 17.43–19.23, and FG-FUZZY-G: 16.02–17.48.

Figure 8 M1 model output maps for (A) ANFIS-SC, (B) ANFIS-GP, (C) ANFIS-FCM and (D) FG.

Figure 9 M2 model output maps for (A) ANFIS-SC, (B) ANFIS-GP, (C) ANFIS-FCM and (D) FG.

Figure 10 M3 model output maps for (A) ANFIS-SC, (B) ANFIS-GP, (C) ANFIS-FCM and (D) FG.

Furthermore, the temperature map helps experts evaluate the temperature in different locations to identify places with maximum or minimum temperatures. For example, in M1, Osmaniye, in M2, Adana, and in M3, Hatay had the highest temperatures in the best FG method. The unique functionality of other GISs in this section helps address defects in the proposed model. Following this, if there is information loss in some areas, the maps use interpolation functions to yield the temperature.

Discussion

As mentioned in the previous section, the ANFIS-GP has memory-related problems when it has much membership functions (more than 3 in this study). In fact, this also changes with respect to input numbers. In this study, 4 inputs were used and, in this case, the use of 4 membership functions causes memory problem. This is due to the fact that ANFIS-GP consider all rules combinations, and it therefore has much more premise parameters, which show the membership functions’ shape and location and consequent parameters that compose the equations of considered rules. However, the main advantage of this method is that it is more flexible because of high number of weights compared to other ANFIS methods, ANFIS-SC and ANFIS-FCM. The main advantages of the ANFIS-SC and ANFIS-FCM methods are the use of less parameters or weights because they use clustering algorithms, and they can therefore optimize rule or membership functions’ numbers. However, this property may cause inappropriate learning of these methods compared to ANFIS-GP.

The FG model was found to be the best method in modelling long-term temperatures, followed by the ANFIS-GP method. The FG also uses all rules combinations similar to ANFIS-GP and therefore, it has much more premise parameters which show the membership functions’ shape and location and consequent parameters. The main advantage of the FG due to its evolutionary training algorithm (genetic algorithm, GA). In ANFIS-GP, gradient descent (GD) algorithm is utilized for the optimization of membership functions’ parameters (premise parameters). The GD can be trapped into local optima, while this does not occur for the GA algorithm.

Conclusions

In this paper, the capabilities of evolutionary fuzzy genetics were compared with three neuro-fuzzy techniques in terms of estimating long-term air temperatures for sustainability. Data from 71 stations from Turkey were used and divided into three equal parts. The applied models were tested using each part. Each method was hence tested three times with all the data. Finally, a GIS was used to produce air temperature maps based on the results of the optimal models. Three main results can be drawn from the application:

i- The evolutionary FG model is superior to the ANFIS-GP, ANFIS-SC, and ANFIS-FCM at modeling long-term air temperatures.

ii- Of the neuro-fuzzy methods, the ANFIS-GP outperformed the ANFIS-SC and ANFIS-FCM.

iii- The accuracy of the best neuro-fuzzy model, the ANFIS-GP, in terms of RMSE, MAE, NSE and R2 increased by 20%, 30%, 4%, and 4%, respectively, using the FG model.

Supplemental Information

Supplemental Information 1 Data and source code

Click here for additional data file.

Additional Information and Declarations

Competing Interests

Author Contributions

Data Availability

The authors declare there are no competing interests.

Abolghasem Sadeghi-Niaraki and Ozgur Kisi conceived and designed the experiments, performed the experiments, analyzed the data, prepared figures and/or tables, and approved the final draft.

Soo-Mi Choi conceived and designed the experiments, authored or reviewed drafts of the paper, project administration, Supervision, Funding acquisition, and approved the final draft.

The following information was supplied regarding data availability:

Data and code are available as Supplementary Files.

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
