# Peer review of "Spatial modeling of long-term air temperatures for sustainability: evolutionary fuzzy approach and neuro-fuzzy methods"

_PeerJ, doi:10.7717/peerj.8882_

## Round 0.1 · original submission · Major Revisions

Both reviewers have made similar comments, so if you address these, the article should be suitable to publish. Please ensure the paper is proof-read before re-submission - PeerJ does not offer language support and editing as part of their normal production process.

You will need to add some more up-to-date literature, add to the discussion and check the Figures and Tables to ensure they are correct and all sources are acknowledged. More care is needed to ensure these are all correct and of publishing quality.



Reviewer 1 ·

Basic reporting

The paper is generally well written, and easy to follow. The methods are reliable, and sound. The results are well presented.

Experimental design

Original primary research within Scope of the journal.
Methods described with sufficient detail and information to replicate.

Validity of the findings

All underlying data have been provided; they are robust, statistically sound, & controlled.

Additional comments

(1) Introduction:
Summary of existing comparisons of the ANN and the three ANFIS methods is not adequate enough. There are many references available:
Prediction of hourly solar radiation in Abu Musa Island using machine learning algorithms;
Modeling daily water temperature for rivers: comparison between adaptive neuro-fuzzy inference systems and artificial neural networks models
….
(2) Four statistics used in the study should be moved to the Methods section rather than in Section 4.
(3) Discussion of the results is not enough. I suggest adding more in-depth analysis in the Section 4 or adding a new section “Discussion”.
(4) Figures:
Figure formats and quality need improvement, such as Fig. 5. Where are the points and lines? Before the final submission, the authors should check carefully about the pdf file, this is the basic requirement. Figs. 6 and 7, can you adjust to make them clear and precise? The same problem for Figs. 9 and 10.
(5) Tables:
The contents in Table 1 didn’t show up completely, please check.
(6) Formulas:
Formulas (7) and (8), please revise the positions of (7) and (8).
(7) Language
The English should be improved. I suggest proof-reading by a native English speaker.

Reviewer 2 ·

Basic reporting

- In general, the article is well written with an appropriate use of scientific terminology.

- However, authors should increase the number of references in the introduction. Between lines 59-67, only old references are cited. It is suggested to add more recent publications when contextualizing the subject of GIS interpolations. For example, Brunetti et al. (2014); Gonzalez-Hidalgo et al. (2015); Serrano-Notivoli (2019); among others.

Brunetti, M. , Maugeri, M. , Nanni, T. , Simolo, C. and Spinoni, J. (2014), High‐resolution temperature climatology for Italy: interpolation method intercomparison. Int. J. Climatol., 34: 1278-1296. doi:10.1002/joc.3764

Gonzalez‐Hidalgo, J. C., Peña‐Angulo, D. , Brunetti, M. and Cortesi, N. (2015), MOTEDAS: a new monthly temperature database for mainland Spain and the trend in temperature (1951–2010). Int. J. Climatol., 35: 4444-4463. doi:10.1002/joc.4298

Serrano-Notivoli, R., Beguería, S., and De Luis, M.: STEAD: A high-resolution daily gridded temperature dataset for Spain, Earth Syst. Sci. Data Discuss., https://doi.org/10.5194/essd-2019-52, in review, 2019.

- Please, in Figure 1, review the source of the image. Who is the provider of the map? What is the meaning of DSI initials in the bottom-left corner?

- Please check the labels of figures 5,6 and 7 because they are misspelling the degrees Celsius symbol: (◦C), not (oC).

Experimental design

The Experimental design has practically no loopholes since the methods used are widely described which allow the readers to understand the whole process.

However, the question arises whether the authors have reviewed the quality of the temperature data. Perhaps they should provide some homogeneity test to guarantee the validity of the data used. For example, a simple method to apply would be the Von Neumann test.

Validity of the findings

the authors provided different metrics to evaluate the interpolation error, thus giving robustness to the results presented.

However, the authors did not offer any kind of discussion that would allow their results to be compared with other similar studies. It is mandatory to add a discussion section to discuss the results obtained.

Additional comments

The authors satisfactorily interpolated the temperature in Turkey, using an advanced and complex methodology.

The article deserves to be published in the journal if the authors introduce the changes and improvements suggested in the review.

---

## Round 0.2 · Minor Revisions

Please address the concerns remaining from reviewer 1. I apologise for the time it has taken to respond to your resubmission.

Reviewer 1 ·

Basic reporting

The paper is generally well written, and easy to follow. The methods are reliable, and sound. The results are well presented. The authors have addressed most of the review comments.

Experimental design

Original primary research within Scope of the journal.
Methods described with sufficient detail and information to replicate.

Validity of the findings

All underlying data have been provided; they are robust, statistically sound, &
controlled.

Additional comments

(1) The authors added more references in the Introduction section as suggested. However, I still have some concerns. In lines 77-83, the authors have already stated the goal and the contents of this study. I am quite puzzled why the last paragraph (Lines 84-91) were placed there. The authors should carefully read the Introduction section again to make it more clear.
(2) Still, figure quality (Figs. 1, 8-10) is not good enough, which needs further improvement.

---

## Round 0.3 · Minor Revisions

Turkey has been spelled as Tukey on lines 62, 65 and 68. As PeerJ does not copy edit you will need to correct this and return.

Line 92 for lats and longs use 39 °N and 35 °E. The minutes and seconds are incorrect (this is to the closest degree. Decimal is 38.9637° N, 35.2433° E)

With references they either all need a comma in the reference between name and year, and references in the same parentheses are separated by a semi-colon or none have commas and references are separated by a comma. At the moment, they are half-and-half

The temperature ranges between lines 309 and 318 need to be separated by n-dashes, especially because some are negative. – - is preferable to --

Please correct these and the paper can be accepted

---

## Round 0.4 · accepted · Accept

Thank you for making those final edits. I hope you were happy with the process.